# Human DDX6 regulates translation and decay of inefficiently translated mRNAs

**Ramona Weber[1,2], Chung-Te Chang[1,3]***

[1]Department of Biochemistry, Max Planck Institute for Developmental Biology, Tübingen, Germany; [2]Institute for Regenerative Medicine (IREM), University of Zurich, Zurich, Switzerland; [3]Institute of Biochemistry and Molecular Biology, National Yang Ming Chiao Tung University, Taipei City, Taiwan

**\*For correspondence:**
chungte.chang@gmail.com

**Competing interest:** The authors declare that no competing interests exist.

**Abstract** Recent findings indicate that the translation elongation rate influences mRNA stability. One of the factors that has been implicated in this link between mRNA decay and translation speed is the yeast DEAD-box helicase Dhh1p. Here, we demonstrated that the human ortholog of Dhh1p, DDX6, triggers the deadenylation-dependent decay of inefficiently translated mRNAs in human cells. DDX6 interacts with the ribosome through the Phe-Asp-Phe (FDF) motif in its RecA2 domain. Furthermore, RecA2-mediated interactions and ATPase activity are both required for DDX6 to destabilize inefficiently translated mRNAs. Using ribosome profiling and RNA sequencing, we identified two classes of endogenous mRNAs that are regulated in a DDX6-dependent manner. The identified targets are either translationally regulated or regulated at the steady-state-level and either exhibit signatures of poor overall translation or of locally reduced ribosome translocation rates. Transferring the identified sequence stretches into a reporter mRNA caused translation- and DDX6-dependent degradation of the reporter mRNA. In summary, these results identify DDX6 as a crucial regulator of mRNA translation and decay triggered by slow ribosome movement and provide insights into the mechanism by which DDX6 destabilizes inefficiently translated mRNAs.

## eLife assessment

This study provides **valuable** findings that improve our understanding of the evolutionary conservation of the role of DDX6 in mRNA decay. The evidence supporting the authors' conclusions is **convincing**. This work will be of interest to molecular, cell biologists and biochemists, especially those studying RNA.

## Introduction

mRNA translation is a highly controlled process that critically determines the mRNA turnover rate. In particular, the elongation rate at which the ribosome translates an mRNA into the nascent peptide chain is thought to affect mRNA stability (*D'Orazio and Green, 2021*, *Morris et al., 2021*). Many mRNA attributes can affect the speed of the translating ribosome, including secondary structure (*Doma and Parker, 2006*), nucleotide and codon composition (*Chaney and Clark, 2015*; *Gardin et al., 2014*; *Letzring et al., 2010*; *Forrest et al., 2020*; *Hia et al., 2019*; *Wu et al., 2019*; *Narula et al., 2019*), tRNA abundance (*Ishimura et al., 2014*; *Novoa and Ribas de Pouplana, 2012*; *Nakahigashi et al., 2014*), the sequence of the nascent peptide and its interaction with the ribosome exit tunnel or other ribosome-associated factors (*Brandman et al., 2012*; *Charneski and Hurst, 2013*; *Kuroha et al., 2010*; *Tanner et al., 2009*; *Lin et al., 2020*; *Buschauer et al., 2020*; *Absmeier et al., 2023*), and damage or improper processing of the mRNA itself (*Brandman and Hegde, 2016*; *Joazeiro, 2019*; *Simms et al., 2017*).

Recent studies have highlighted the important roles of codon optimality, rare codons, and stalling sequences in protein synthesis. Codon optimality refers to the preferential usage of certain codons over others during translation, typically based on their efficiency and accuracy. Rare codons are codons that are infrequently used in a particular organism's genome. These rare codons may be recognized by tRNAs with lower abundance, leading to slower or less accurate protein production. Stalling sequences, on the other hand, are specific mRNA sequences that can cause the ribosome to pause or stall during translation, which can occur due to factors such as secondary structures in the mRNA or interactions with specific proteins (*Hanson and Coller, 2018*).

A number of studies have found that codon usage bias affects translation elongation rates and protein synthesis efficiency, emphasizing the importance of synonymous codon changes on translation speed and accuracy (*Yu et al., 2015*; *Gardin et al., 2014*). Moreover, rare codon usage also influences protein folding dynamics, highlighting the intricate relationship between codon optimality and protein structure formation (*Charneski and Hurst, 2013*). It is evident that the diverse mechanisms underlying stalling sequences, including interactions with chaperone proteins and ribosome-associated factors, provide insights into the molecular basis of translational stalling and its impact on protein biogenesis (*Sitron and Brandman, 2020*). Furthermore, it is now understood that recognition of slow ribosome dynamics by the mRNA decay machinery can trigger mRNA degradation, not only during ribosome-associated quality control (RQC), but also in general translation regulation (*Buskirk and Green, 2017*; *Buschauer et al., 2020*; *Radhakrishnan et al., 2016*).

In eukaryotes, general mRNA decay is initiated by a shortening of the poly(A) tail mediated by the CCR4-NOT complex with its catalytic subunits CNOT7 and CNOT8, followed by 5′-to-3′ mRNA decay, where the mRNA is decapped by the DCP1/DCP2 decapping complex and then exonucleolytically degraded by the exonuclease XRN1 (*Mugridge et al., 2018*). Alternatively, the exosome can degrade deadenylated mRNAs in the 3′-to-5′ direction (*Robinson et al., 2015*; *Łabno et al., 2016*). There are also more specialized RQC mechanisms, such as no-go decay and non-stop decay, which remove aberrant mRNAs often by endonucleolytic cleavage (*Simms et al., 2017*; *Inada, 2017*).

As mentioned above, mRNAs with slow translation rates due to the presence of non-optimal codons are regulated by translation-dependent decay (*Radhakrishnan and Green, 2016*; *Richter and Coller, 2015*; *Presnyak et al., 2015*; *Boël et al., 2016*; *Herrick et al., 1990*; *Narula et al., 2019*; *Hia et al., 2019*; *Wu et al., 2019*; *Forrest et al., 2020*). One factor implicated in translation-dependent decay in yeast is Dhh1p, the ortholog of human DDX6 (*Sweet et al., 2012*; *Radhakrishnan et al., 2016*). DDX6 belongs to the conserved DEAD-box ATPase/helicase family, which plays a central role in cytoplasmic mRNA regulation, including processing body (P-body) assembly, mRNA decapping, and translational repression (*Ostareck et al., 2014*). DDX6 is composed of two globular RecA-like domains connected by a flexible linker. The helicase motifs that are crucial to the ATPase and RNA binding activities are located in the DDX6-RecA1 domain. The DDX6-RecA2 domain mediates binding to the translational regulators 4E-T and GIGYF1/2, the decapping activators PATL1, EDC3, LSM14A, and to the NOT1 subunit of the CCR4-NOT complex, thereby providing a link between mRNA translational repression and 5′-to-3′ mRNA decay by deadenylation and decapping (*Mathys et al., 2014*; *Chen et al., 2014*; *Peter et al., 2019*).

Despite its well-known involvement in translation-coupled mRNA turnover in yeast, the target mRNAs and the molecular mechanism of mRNA recognition by mammalian DDX6 remain elusive. In this study, we investigated the role of human DDX6 in translation-dependent mRNA decay. Comparing the stability of a reporter mRNA containing a translational stalling stretch in both HEK293T wild-type (WT) and DDX6 knockout (KO) cells revealed that DDX6 destabilizes inefficiently translated reporter mRNA in human cells, in a similar manner as has been observed in yeast (*Radhakrishnan et al., 2016*), with this destabilization being mediated through initiation of the 5'-to-3' mRNA decay pathway. Furthermore, we found that the DDX6 RecA2 domain interacts with ribosomal proteins and provides evidence that RecA2-mediated interactions and DDX6-ATPase activity are both necessary for translation-dependent mRNA decay. To identify the set of inefficiently translated transcripts that are naturally targeted by DDX6 we took an unbiased approach and compared HEK293T WT and DDX6 KO cells by ribosome profiling and RNA sequencing analysis. This approach identified a set of mRNAs that are translationally regulated by DDX6, many encoding zinc finger transcription factors, and have low overall codon optimality and thus slow translation elongation dynamics across the entire open reading frame. We also identified a second group of mRNAs with locally decreased elongation

dynamics targeted by DDX6 for decay. Interestingly, the second group of mRNAs included transcripts with CAG repeats, which have recently been linked to slow ribosome dynamics (*Aviner et al., 2022*). These findings suggest that loss of DDX6 could disrupt co-translational proteostasis of factors associated with neurodegenerative diseases, leading to the characteristic formation of intracellular aggregates. Through an mRNA reporter approach, we also found that the newly identified mRNA regions promote destabilization in a translation- and DDX6-dependent manner. Together, our study establishes that the human RNA helicase DDX6 plays a critical role in mediating translation efficiency and mRNA stability in human cells.

## Results
### Codon composition affects mRNA stability in human cells

Codon composition was recently identified as a strong determinant of mRNA stability in yeast, *Drosophila,* and different vertebrate species (*Presnyak et al., 2015*; *Bazzini et al., 2016*; *Mishima and Tomari, 2017*). Emerging evidence indicates that the translation elongation rate is monitored by the ribosome and ribosome-associated factors, triggering mRNA decay when the ribosome slows down (*Hu et al., 2009*; *Sweet et al., 2012*; *Presnyak et al., 2015*; *Radhakrishnan and Green, 2016*; *Forrest et al., 2020*; *Hia et al., 2019*; *Narula et al., 2019*; *Wu et al., 2019*). Codon composition can impact protein synthesis through distinct concepts such as codon optimality, rare codons, and stalling sequences (*Hanson and Coller, 2018*). Among these, rare codons - specific codons that are not frequently used in certain organisms - have been prominently studied in yeast models, though fewer studies have examined their effects in human cells. To investigate the effects of rare codons in more detail in human cells, we created a reporter construct containing a *Renilla* luciferase open reading frame, wherein the final 30 codons were replaced by synonymous but rarely used codons (*Figure 1A*), guided by the codon rarely utilized scores sourced from the Codon-Usage Database (https://www.kazusa.or.jp/codon/). This change was designed to reduce the translation speed without altering the resulting amino acid sequence (*Hussmann et al., 2015*; *Weinberg et al., 2016*; *Nakahigashi et al., 2014*; *Gardin et al., 2014*; *Yu et al., 2015*; *Letzring et al., 2010*). To find out whether the expected decrease in ribosome translocation rates indeed alters mRNA stability, we measured the half-life of the reporter mRNA in human HEK293T cells. As expected, introducing the rare codon stretch into the *Renilla* luciferase open reading frame significantly reduced the half-life of the translated mRNA from >10 hr to 8.0±4.4 hr (*Figure 1B and C*). This result confirms that codon composition directly affects mRNA stability in human cells (*Chen and Coller, 2016*), and is in line with previous reports describing a correlation between codon optimality and mRNA stability in various mammalian cell systems (*Bazzini et al., 2016*; *Radhakrishnan and Green, 2016*; *Schwanhäusser et al., 2011*; *Goodarzi et al., 2016*).

In *Saccharomyces cerevisiae* (*S.c.*) the DEAD-box RNA helicase Dhh1p (ortholog of human DDX6) has been shown to act as a sensor of codon optimality (*Sweet et al., 2012*; *Radhakrishnan et al., 2016*). To test whether its human ortholog DDX6 is needed to trigger the decay of not optimally translated mRNAs, we created HEK293T DDX6 KO cells using CRISPR/Cas9-mediated gene editing. Loss of DDX6 protein expression was verified by western blotting and sequencing of the genomic target site (*Figure 1—figure supplement 1A,B*). Remarkably, the loss of DDX6 significantly stabilized the non-optimal codon-containing reporter, resulting in a measured half-life comparable to that of the respective control reporter (>10 hr vs 8.0±4.4 hr) (*Figure 1B and C* right panel). These results indicate that DDX6 acts as a sensor of codon optimality and that this function is evolutionarily conserved from yeast to mammals.

In budding and fission yeast, co-translational degradation of mRNAs is mediated by the 5′-to-3′ decay pathway (*Hu et al., 2009*; *Pelechano et al., 2015*). It has also been reported that maternal codon-mediated mRNA clearance in zebrafish is triggered by co-translational CCR4-NOT-dependent deadenylation (*Mishima and Tomari, 2017*). To test whether the observed degradation of the rare-codon reporter is dependent on CCR4-NOT, we measured its half-life in cells overexpressing a catalytically inactive POP2 deadenylase mutant (POP2 DE-AA). Blocking CCR4-NOT dependent deadenylation indeed resulted in marked stabilization of the reporter mRNA, with half-lives increasing from 5.5±2.8 hr to >10 hr (*Figure 1D and E*). This result shows that in human cells, mRNA decay triggered

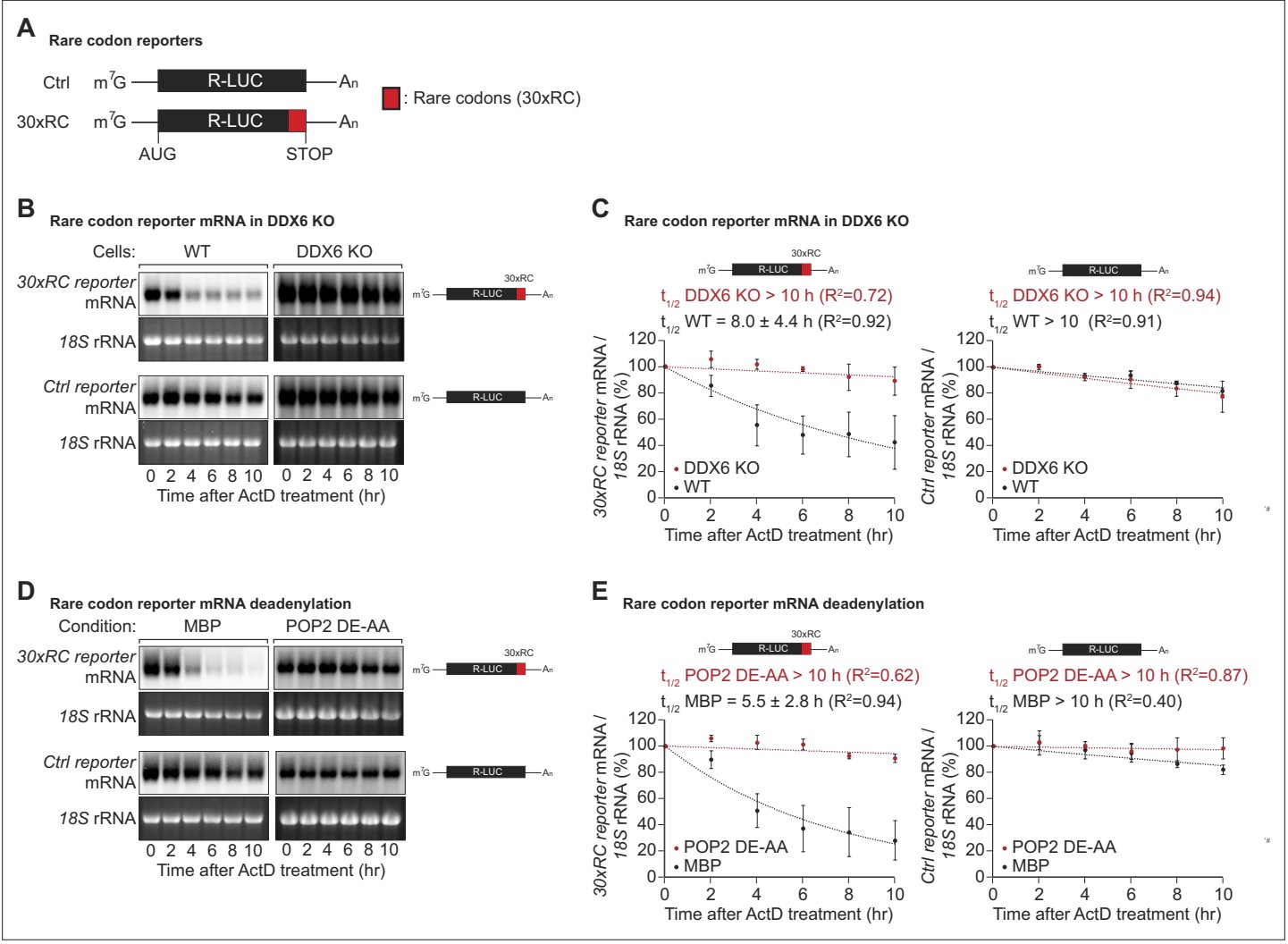

**Figure 1.** DDX6 functions as a sensor of rare codon-triggered mRNA decay in human cells. (**A**) Schematic representation of the reporters used in panels (**B, D**). (**B**) Wild-type (WT) and DDX6 KO HEK293T cells were transfected with indicated reporter plasmids. After 48 hr, cells were treated with actinomycin D (ActD) and harvested at the indicated time points. Reporter mRNA levels were analyzed by northern blotting. *18* S rRNA ethidium bromide staining shows equal loading. (**C**) Relative reporter mRNA levels from panel B at time point zero (before ActD addition) were defined as 100%. Relative reporter mRNA levels were plotted as a function of time. Circles represent the mean value and error bars the standard deviation (SD) (n=3). The decay curves were fitted to an exponential decay with a single component (dotted lines). $R^2$ values are indicated for each curve. The half-life of each mRNA in WT and DDX6 KO cells is represented as the mean ± SD. (**D**) HEK293T cells were transfected with MBP or POP2 dominant negative mutant (POP2 DE-AA) and indicated reporter plasmids. After 48 hr, cells were treated with ActD and harvested at the indicated time points. Reporter mRNA levels were analyzed by northern blotting. *18* S rRNA ethidium bromide staining shows equal loading. (**E**) Relative reporter mRNA levels from panel D at time point zero (before ActD addition) were defined as 100%. Relative reporter mRNA levels were plotted as a function of time. Circles represent the mean value and error bars the SD (n=3). The decay curves were fitted to an exponential decay with a single component (dotted lines). $R^2$ values are indicated for each curve. The half-life of each reporter mRNA in WT and POP2 DE-AA overexpressing cells is represented as the mean ± SD.

The online version of this article includes the following source data and figure supplement(s) for figure 1:

**Source data 1.** Original file for the northern blot analysis in *Figure 1B*.

**Source data 2.** PDF containing *Figure 1B* and original scans of the relevant northern blot analysis with highlighted bands and sample labels.

**Source data 3.** Original file for the northern blot analysis in *Figure 1D*.

**Source data 4.** PDF containing *Figure 1D* and original scans of the relevant northern blot analysis with highlighted bands and sample labels.

**Figure supplement 1.** Characterization of HEK293T DDX6 KO cells.

**Figure supplement 1—source data 1.** Original file for the western blot analysis in *Figure 1—figure supplement 1A*.

**Figure supplement 1—source data 2.** PDF containing *Figure 1—figure supplement 1A* and original scans of the relevant western blot analysis with highlighted bands and sample labels.

*Figure 1 continued on next page*

*Figure 1 continued*

**Figure supplement 1—source data 3.** Original file for the northern blot analysis in *Figure 1—figure supplement 1C*.

**Figure supplement 1—source data 4.** PDF containing *Figure 1—figure supplement 1C* and original scans of the relevant northern blot analysis with highlighted bands and sample labels.

by rare codon-related slow ribosome movement is initiated by CCR4-NOT-dependent deadenylation of the target mRNA.

Previous studies have shown that both the helicase DDX6 and CCR4-NOT-dependent deadenylation were required for mRNA decay triggered by a slowed ribosome. However, whether DDX6 directly regulates the deadenylation activity of CCR4-NOT was not investigated. Given that the CCR4-NOT complex is known to interact with the empty ribosome E-site in both yeast and humans (*Absmeier et al., 2023*; *Buschauer et al., 2020*), the potential impact of DDX6 on CCR4-NOT function was an important open question. By using tethering assays with CNOT3, a major subunit of the CCR4-NOT complex known to interact with the empty E site of the ribosome (*Absmeier et al., 2023*), we have demonstrated that knockout of DDX6 does not affect the deadenylation activity of the complex (*Figure 1—figure supplement 1C*). This suggests that DDX6 participates in the mRNA decay process triggered by a slowed ribosome through a different mechanism, rather than by modulating CCR4-NOT activity. This additional data helps provide a more complete understanding of the mechanisms underlying mRNA decay in the context of a slowed ribosome.

## DDX6 associates with the ribosome components

To sense translation speed, it has been proposed that DDX6 may engage in physical interactions with the ribosome, and yeast Dhh1p has indeed been found to co-purify with ribosomal components (*Radhakrishnan et al., 2016*; *Sweet et al., 2012*; *Drummond et al., 2011*). To test whether human DDX6 interacts with the ribosome, we performed in vitro pulldown assays with recombinant DDX6 purified from *Escherichia coli* (*E. coli*) lysates, and endogenous ribosomal complex components purified from HEK293T cells. Streptavidin-tagged DDX6 pulled down several ribosomal protein components (labeled with a red dot), whereas no such interaction was observed in the corresponding streptavidin-MBP pulldown (*Figure 2A*, lane 4 vs 6). Despite RNase I treatment to eliminate potential RNA bridges, it remains possible that factors associated with the ribosome and interacting with DDX6 persist even after purification from the sucrose cushion. These results, together with previous studies demonstrating the involvement of DDX6 orthologs in ribosome-associated processes, suggest that the ability of DDX6 orthologs to associate with ribosomes is a conserved feature across different species.

To determine precisely which part of DDX6 contacts the ribosome, we performed co-immunoprecipitations with DDX6 protein fragments in HEK293T cells. Since we did not know which ribosomal component was mediating the interaction with DDX6, immunoprecipitations were performed in the presence of RNaseI. RNaseI degrades unprotected mRNAs but leaves the ribosome structure and ribosome-protected RNA regions intact (*Ingolia et al., 2012*). We found that under these conditions, overexpressed full-length GFP-tagged DDX6 and the C-terminal RecA2 domain co-immunoprecipitated with both the HA-tagged large ribosomal subunit protein RPL22 and the small ribosomal subunit component RPS3A in human cells. In contrast, the N-terminal RecA1 domain did not interact with either subunit (*Figure 2B*). The RecA2 domain of DDX6 mediates multiple protein interactions. It binds the Mid domain of eIF4G (MIF4G) of NOT1 which stimulates the DDX6-ATPase activity by stabilizing it in an activated conformation (*Mathys et al., 2014*; *Chen et al., 2014*). In addition to providing a link to the CCR4-NOT deadenylase complex, the RecA2 domain of DDX6 also mediates interactions with other regulatory proteins e.g., EDC3, LSM14A, PATL1, 4E-T, and GIGYF1/2. These proteins interact with DDX6 RecA2 through their Phe-Asp-Phe (FDF) motifs in a mutually exclusive manner, thereby either facilitating mRNA decay by linking the deadenylation machinery to the core decapping complex or inducing translational repression (*Fromm et al., 2012*; *Tritschler et al., 2008*; *Nissan et al., 2010*; *Sharif et al., 2013*; *Tritschler et al., 2009*; *Brandmann et al., 2018*; *Ozgur et al., 2015a*; *Peter et al., 2019*).

To obtain further insights into how DDX6 interacts with the ribosome, we compared the interactions of different DDX6 mutants with the large ribosomal subunit protein RPL22 in HEK293T cells. In particular, we used a DDX6 mutant (Mut1; R94E+F101 D+Q322 A+N324 A+R375 A) that does

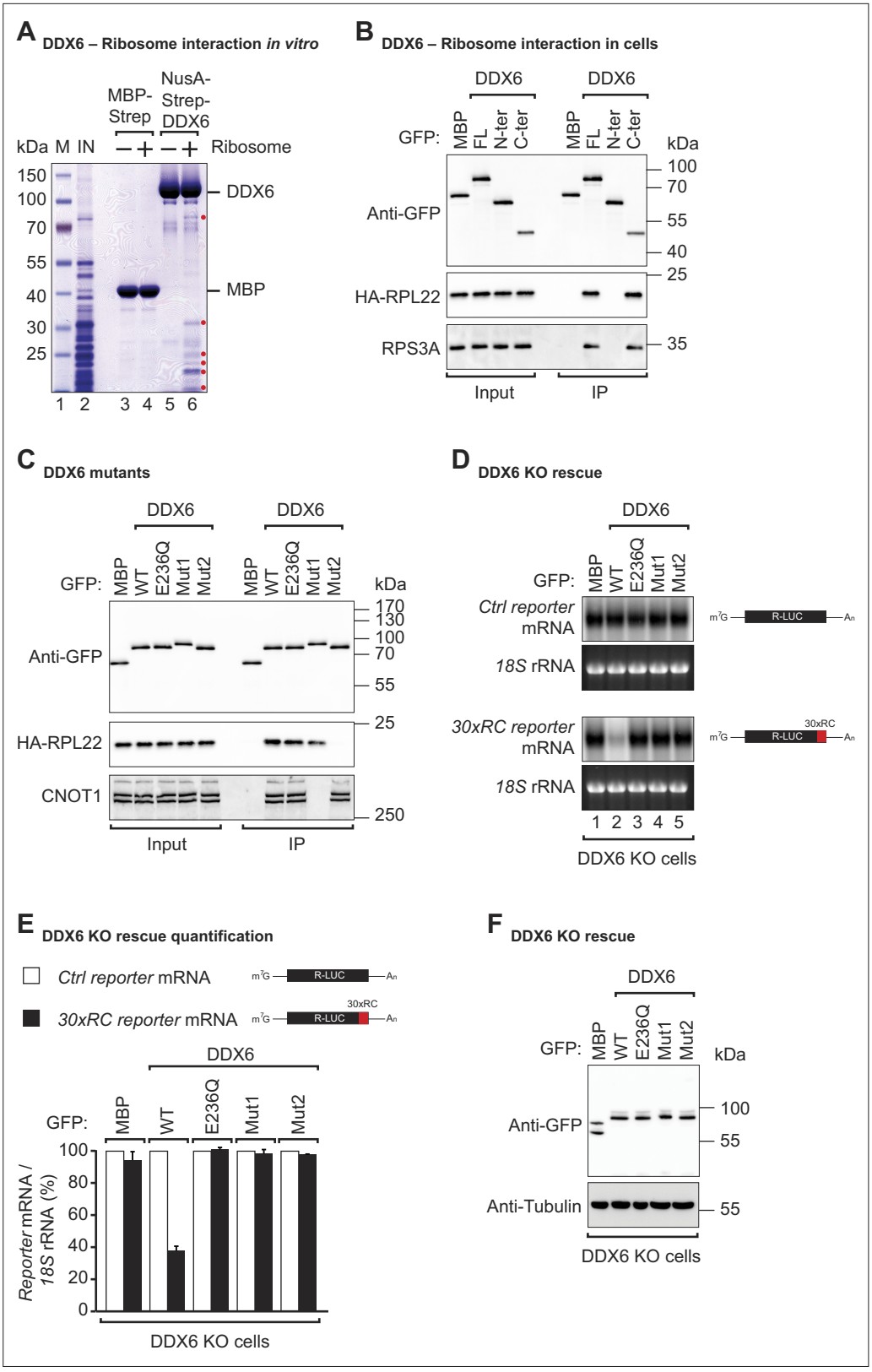

**Figure 2.** DDX6 interacts with ribosomal proteins in human cells. (**A**) The interaction between the recombinant NusA-Strep-DDX6 and purified human ribosomal proteins was analyzed by SDS-PAGE and stained with Coomassie blue. Input lysate (1%) and bound fractions (20%) were loaded. (**B**) Western blot showing the interaction between GFP-tagged DDX6 full-length/N-ter/C-ter with HA-tagged RPL22 and endogenous RPS3A in human HEK293T

*Figure 2 continued on next page*

*Figure 2 continued*

cells. GFP-tagged MBP served as a negative control. For the GFP-tagged proteins, the HA-tagged RPL22, and the endogenous RPS3A, 1% of the input and 20% of the immunoprecipitate were loaded. N-ter: N-terminus; C-ter: C-terminus. (**C**) Immunoprecipitation assay showing the interaction of GFP-tagged DDX6 (wild-type or the indicated mutants) with HA-tagged RPL22 or endogenous CNOT1 in HEK293T cells. Samples were analyzed as described in B. (**D**) DDX6 KO HEK293T cells were transfected with the control *Renilla* luciferase (R-LUC) reporter or a reporter containing 30 x rare codons and GFP-tagged DDX6 wild-type or mutants. After treating cells with ActD for 8 hr R-LUC mRNA levels were analyzed by northern blotting. 18 S rRNA ethidium bromide staining shows equal loading. (**E**) Relative control reporter mRNA levels from panel D were defined as 100%. Relative 30 x rare codon reporter mRNA levels were plotted. Bars represent the mean value and error bars the standard deviation (n=3). (**F**) Immunoblot illustrating the expression of proteins used in the assay shown in panel D.

The online version of this article includes the following source data and figure supplement(s) for figure 2:

**Source data 1.** Original file for the SDS-PAGE in *Figure 2A*.

**Source data 2.** PDF containing *Figure 2A* and original scans of the relevant SDS-PAGE with highlighted bands and sample labels.

**Source data 3.** Original file for the western blot in *Figure 2B*.

**Source data 4.** PDF containing *Figure 2B* and original scans of the relevant western blot analysis with highlighted bands and sample labels.

**Source data 5.** Original file for the western blot in *Figure 2C*.

**Source data 6.** PDF containing *Figure 2C* and original scans of the relevant western blot analysis with highlighted bands and sample labels.

**Source data 7.** Original file for the northern blot in *Figure 2D*.

**Source data 8.** PDF containing *Figure 2D* and original scans of the relevant northern blot analysis with highlighted bands and sample labels.

**Source data 9.** Original file for the western blot in *Figure 2F*.

**Source data 10.** PDF containing *Figure 2F* and original scans of the relevant western blot analysis with highlighted bands and sample labels.

**Figure supplement 1.** Multidimensional scaling analysis of Ribo-Seq and RNA-Seq and the ribosome footprints on mRNA read distribution in DDX6 KO Cells.

not interact with NOT1 but still interacts with the FDF-motif proteins. We also tested a mutant with substitutions on the FDF-binding surface (Mut2; Q209A+H312 A+T316 A+R320 A+R335 A+K341 A+K342 A). These mutations prevent binding to regulatory factors such as EDC3, LSM14A, PATL1, 4E-T, and GIGYF1/2, thereby breaking the link with the decapping and translation repression machinery (*Tritschler et al., 2009*; *Kuzuoğlu-Öztürk et al., 2016*; *Ozgur et al., 2015a*; *Peter et al., 2019*). As a control, we also tested a DDX6 DEAD-box mutant (E236Q). Interestingly, the mutation of the FDF-binding surface abolished DDX6's interaction with the ribosomal protein RPL22, whereas mutations disrupting the interaction with NOT1 and mutation of the DEAD-box motif did not interfere with ribosome binding (*Figure 2C*). The DDX6 mutant with a modified FDF motif (Mut2) still interacted with NOT1, indicating that protein folding was not compromised (*Figure 2C*). In summary, these results demonstrate that DDX6 interacts with the ribosome via the same region that provides a link to the decapping and translation repression machinery.

DDX6 belongs to the DEAD-box family of RNA-dependent ATPases/helicases (*Ozgur et al., 2015b*; *Presnyak and Coller, 2013*; *Russell et al., 2013*). To test whether the ATPase activity is required to promote the decay of rare-codon reporter mRNA, we complemented HEK293T DDX6 KO cells with either wild-type (WT) or DEAD-box mutant (E236Q) DDX6. Interestingly, loss of the ATPase activity completely abolished DDX6's ability to induce a decay of the rare-codon reporter mRNA (*Figure 2D* lane 2 vs lane 3 and *Figure 2E*). Sweet et al. found similarly that in yeast, Dhh1p interaction with polysomes following translation inhibition was dependent on its ATPase activity *Sweet et al., 2012*.

To test whether the interactions mediated by the DDX6 RecA2 domain are also necessary for stalling mediated decay, we complemented DDX6 KO HEK293T cells with either the NOT1 binding mutant (Mut1) or FDF-pocket mutant (Mut2) of DDX6 and examined their effect on the stability of the non-optimal codon reporter. Stalling-mediated mRNA decay was only observed in the presence of WT DDX6 (*Figure 2D* lane 2 and *Figure 2E*). Loss of contact with NOT1 (Mut1) or of FDF-mediated

interactions (Mut2) rendered DDX6 inactive (*Figure 2D* lanes 4, 5, and *Figure 2E*). All proteins were expressed at equivalent levels (*Figure 2F*). These data show that DDX6 interacts with the ribosome via its C-terminal RecA2 domain and that all domains and its catalytic activity are required for DDX6 to mediate mRNA decay of a poorly translated reporter.

## Identification of endogenous mRNAs targeted by DDX6 for decay because of inefficient translation elongation

To determine which cellular mRNAs are naturally sensed by DDX6 and targeted for decay, we performed ribosome profiling combined with RNA sequencing (RNA-seq) in HEK293T WT and DDX6 KO cells. Ribosome profiling involves high-throughput sequencing of the RNA footprints of polyribosomes subjected to in vitro RNA digestion (*Ingolia et al., 2009*; *McGlincy and Ingolia, 2017*). This technique can thus reveal genome-wide ribosome occupancy with nucleotide resolution, providing insights into the number and position of ribosomes bound throughout the transcriptome (*Ingolia et al., 2012*). While the overall ribosome occupancy on a given transcript is a proxy for protein synthesis levels, the distribution of ribosome footprints within a transcript is a measure of local ribosome speed. In particular, ribosome footprints tend to accumulate at sites with reduced ribosome speed (*Ingolia et al., 2011*) and can, therefore, be used to study ribosome pausing or stalling (*Li et al., 2012*; *Guydosh and Green, 2014*; *Gamble et al., 2016*). To investigate a possible link between locally increased ribosome occupancy, indicative of slow ribosome elongation, and DDX6-dependent mRNA decay, we compared the RNA-seq and ribosome profiling data of HEK293T WT and DDX6 KO cells. Experimental replicates were comparable as they clustered together (*Figure 2—figure supplement 1A,B*) and reduced DDX6 mRNA expression and translation further validated the HEK293T DDX6 KO cell line (*Figure 2—figure supplement 1C*). Comparison of the RNA-seq datasets revealed 1707 mRNAs specifically upregulated in the absence of DDX6 (FDR <0.005; *Figure 3A and C*). Of those, 298 transcripts were upregulated more than twofold (logFC >1). Gene ontology analysis revealed enrichment for transcripts coding for proteins that localize in the extracellular space, to membranes or are involved in signaling or cell adhesion (*Figure 3E*). On the other hand, 1484 transcripts were significantly downregulated (FDR <0.005; *Figure 3A and C*), 156 more than twofold (logFC < –1). In addition, comparing the ribosome profiling and RNA-seq data allowed us to identify changes in translation efficiency (TE). Remarkably, 260 transcripts were translationally upregulated in DDX6 KO cells (FDR <0.005; *Figure 3B and D*), 152 more than twofold (logFC >1). Functional characterization of these target genes revealed a strong predominance of genes encoding zinc finger transcription factors (89 of the 260 translationally upregulated genes; 34%; FDR = 2.5E-46) (*Figure 3B and D*). Accordingly, GO-analysis showed an enrichment for DNA binding proteins and factors involved in transcription (*Figure 3F*). In contrast, only 38 transcripts showed significant downregulation in TE (FDR <0.005; *Figure 3B and D*), 17 more than twofold (logFC < –1), with no functional association as measured by GO analysis. These results suggest that the loss of DDX6 has a profound effect on the translation efficiency of a specific class of mRNAs and led us to investigate the common molecular features of the 260 mRNA targets translationally repressed by DDX6. Comparison of the 260 upregulated mRNAs in DDX6 KO cells revealed that these mRNAs have significantly lower coding sequence (CDS) GC content than the rest of the mRNAs expressed in the same cells (*Figure 3—figure supplement 1A*). Furthermore, DDX6-regulated CDSes appear to be longer than unaffected mRNA CDSes (*Figure 3—figure supplement 1B*). Interestingly, in mammals, low codon GC content is correlated with low codon optimality (*Plotkin and Kudla, 2011*). Thus, DDX6 may be involved in the regulation of the translation efficiency of mRNAs with low codon optimality across the whole transcript CDS and thus low transcript stability (*Forrest et al., 2020*; *Hia et al., 2019*; *Wu et al., 2019*; *Narula et al., 2019*). To this end we compared DDX6 translationally regulated mRNAs with a transcriptome-wide prediction analysis of mRNA stabilities based on the codon composition (*Diez et al., 2022*) and found that mRNAs translationally regulated by DDX6 (TE up in DDX6 KO cells) are associated with a substantially lower predicted mRNA stability score (*Figure 3G*). In agreement with our data that many transcripts with increased TE in DDX6 KO cells encode zinc finger transcription factors, Diez et al. also report that mRNAs encoding zinc finger transcription factors collectively show a lower stability score than the average transcriptome. These findings suggest that DDX6 translationally regulates a specific group of mRNAs with low codon optimality and, therefore, low stability. Interestingly, DDX6-regulated mRNAs were not found to have a significantly lower TE than the other mRNAs expressed in the cells;

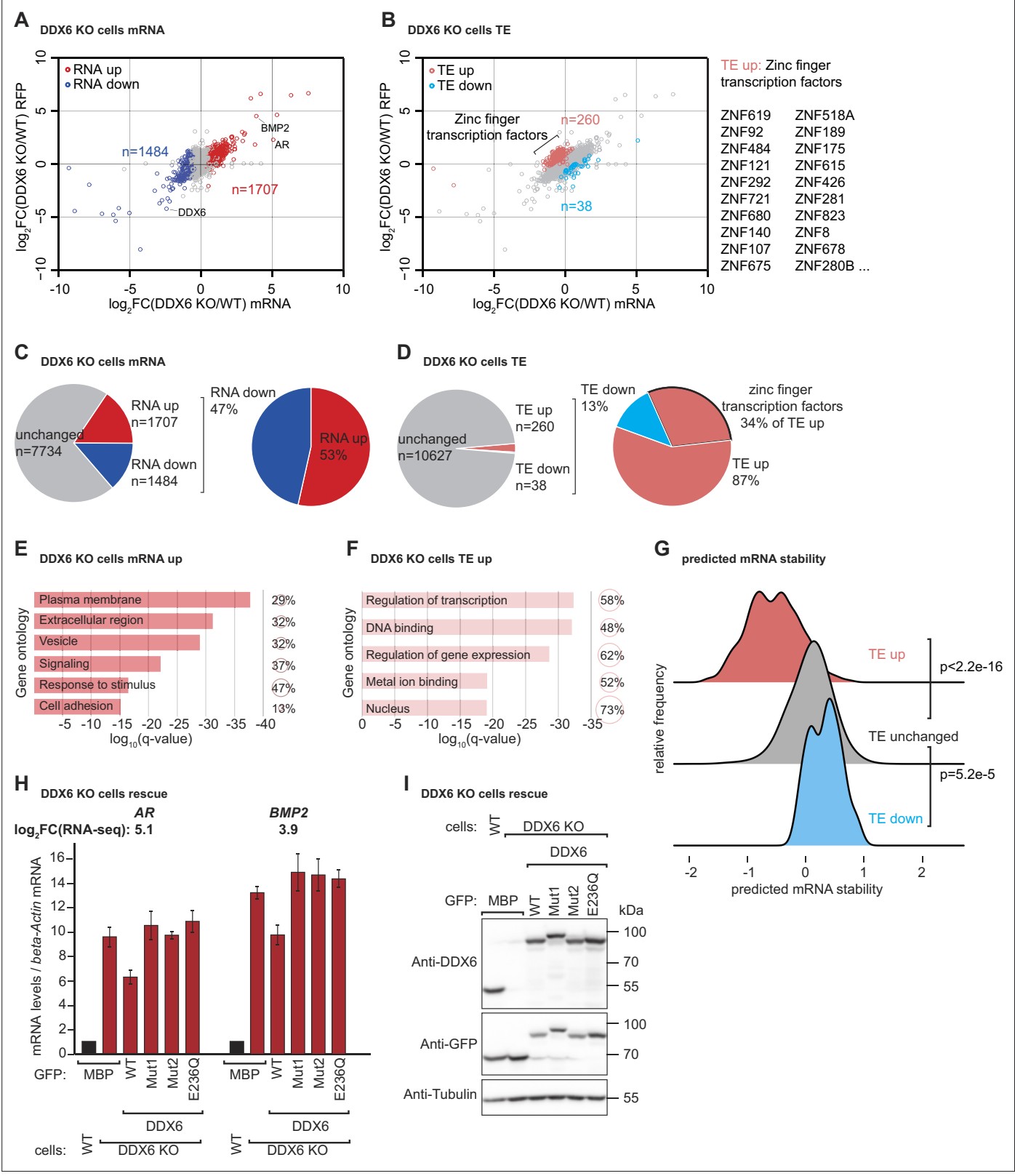

**Figure 3.** DDX6 controls mRNA abundance and translational efficiency in human cells. (**A**) Comparative analysis of translational efficiency (TE) in wild-type (WT) HEK293T and DDX6 KO cells. Genes with significantly (FDR <0.005) increased (n=1707 genes) and decreased (n=1484 genes) mRNA abundance are colored in red and blue, respectively. (**B**) Comparative analysis of TE in WT HEK293T and DDX6 KO cells. Genes with significantly (FDR <0.005) increased (n=260 genes) and decreased (n=38 genes) TE are highlighted in salmon and cyan, respectively. The top 20 (total 89) of translationally

*Figure 3 continued on next page*

*Figure 3 continued*

upregulated zinc finger transcription factors in DDX6 KO cells are highlighted. (**C**) Pie charts indicating the fractions and absolute numbers of significantly (FDR <0.005) differentially expressed mRNAs in HEK293T WT and DDX6 KO cells as determined by RNA-seq. (**D**) Pie charts indicating the fractions and absolute numbers of significantly (FDR <0.005) differentially translated mRNAs in HEK293T WT and DDX6 KO cells as determined by Ribo-seq/RNA-seq. (**E**) Gene ontology of the biological processes associated with upregulated transcripts in DDX6 KO cells. Bar graph shows $\log_{10}$ q-values for each overrepresented category. Values and circles indicate the % of genes within each category. (**F**) Gene ontology of the biological processes associated with translationally upregulated transcripts in DDX6 KO cells. Bar graph shows $\log_{10}$ q-values for each overrepresented category. Values and circles indicate the % of genes within each category. (**G**) Ridgeline plots of predicted mRNA stability (*Diez et al., 2022*) of translationally upregulated unchanged/downregulated transcripts in DDX6 KO cells. Statistical significance was calculated with the one-sided Wilcoxon rank sum test. (**H**) qPCR analysis of *AR* and *BMP2* mRNA levels in HEK293T WT and DDX6 KO and rescued with GFP-tagged DDX6 (wild-type or the indicated mutants). $\log_2 FC$ values for each transcript as determined by the RNA-seq experiments are indicated. (**I**) Immunoblot depicting the expression of proteins used in the assay shown in panel G.

The online version of this article includes the following source data and figure supplement(s) for figure 3:

**Source data 1.** Original file for the westhern blot in *Figure 3I*.

**Source data 2.** PDF containing *Figure 3I* and original scans of the relevant western blot analysis with highlighted bands and sample labels.

**Figure supplement 1.** Characterization of DDX6 target mRNAs.

**Figure supplement 2.** Identification of DDX6 target mRNAs.

**Figure supplement 3.** Validation of DDX6 target mRNAs.

however, loss of DDX6 significantly increased the TE of this subset of mRNAs above the overall TE (*Figure 3—figure supplement 1C*).

In addition to the unexpected finding that DDX6 translationally represses a specific group of endogenous transcripts, we reasoned that we should also find potential targets of DDX6-mediated mRNA decay in the group of mRNAs that are more abundant in the absence of DDX6. Based on our previous observations that DDX6 mediates decay of inefficiently translated mRNAs, we further reasoned that we could use the ribosome profiling data to identify upregulated mRNAs with locally increased ribosome occupancy, indicative of slow elongation rates (overview *Figure 3—figure supplement 2A*). We manually screened the list of transcripts with logFC >1 and a statistically significant change (FDR <0.005) in the RNA-seq experiment (298 transcripts) for the presence of a local increase of ribosome footprints as measured by the coupled ribosome profiling experiment and found 35 genes that matched these criteria (*Figure 3—figure supplement 2B–G* and Material and methods). It is worth mentioning that one of the most highly upregulated mRNAs in DDX6 KO cells is the androgen receptor (*AR*) transcript containing CAG repeats, coding for a stretch of glutamine repeats (poly Q). The repetitive nature of this sequence makes it difficult to determine whether inefficient translation elongation occurs during decoding. However, it is interesting to note that a recent preprint suggests that CAG expansions can cause altered elongation kinetics that can lead to CAG expansion diseases like Huntington's disease (*Aviner et al., 2022*). Repetitive codon sequences potentially lead to ribosome stalling because of limited tRNA availability (*Dana and Tuller, 2014*; *Ishimura et al., 2014*; *Novoa and Ribas de Pouplana, 2012*). Given the repetitive nature of the transcript, we hypothesized that *AR* mRNA could be a direct DDX6 target. We then used quantitative PCR (qPCR) to measure the levels of selected transcripts in WT, DDX6 KO cells, and DDX6 KO cells expressing either WT or mutant DDX6 exogenously. DDX6 dependency was detected in 6 of the 9 tested mRNAs (*Figure 3H* and *Figure 3—figure supplement 3A AR*, *BMP2*, *LGALS1*, *DLX5*, *ENO2*, *PSMB9*; see also changes in mRNA abundance as determined by RNA-seq; note that especially for *AR* and *ENO2* the changes in mRNA abundance appear to be underestimated by the qPCR analysis compared to the RNA-seq quantification which can possibly arise due to differences in normalization strategies of the different technologies or exact sample status). The following three potential DDX6 targets were not confirmed: *C1QTNF4*, *CALCB*, and *PCK1*. We observed that DDX6 KO cells had higher levels of target mRNAs compared to WT HEK293T cells, as also measured by the RNA-seq experiments, and were partially restored back to WT levels by expressing exogenous DDX6. The partial rescue might be explained by uneven transfection efficiency and/or additional time required to reach base line mRNA levels. It should be noted that none of the mutant forms of DDX6 were able to restore target mRNA levels, consistent with the previous reporter experiments. All proteins were expressed at equivalent levels (*Figure 3I*). These results strongly suggest that DDX6 plays a direct role in the regulation of target mRNA levels in HEK293T cells.

## mRNA sequences with increased ribosome occupancy reduce reporter mRNA stability

To test whether the identified translational pause sites in some of the endogenous DDX6-targeted mRNAs contribute to mRNA destabilization, we created different reporter constructs in which the putative ribosomal stalling region (RSR) was inserted downstream of the *Renilla* luciferase open reading frame, either preceding or following a stop-codon; R-LUC-RSR and R-LUC-stop-RSR, respectively (*Figure 4A*). These constructs were designed to differentiate translation-dependent regulation from translation-independent effects, triggered for example by binding of trans-regulatory factors (e.g. RNA-binding proteins or miRNAs) to the introduced sequence. We tested two of the sequences identified in our screen for DDX6-destabilized mRNAs: first, a stretch containing 23 'CAG' codon repeats (encoding 23 x Q) mimicking the X-chromosome encoded AR gene. For the second sequence, we chose a 30 nt stretch of the bone morphogenetic protein 2 (BMP2) gene encoded on chromosome 20, which codes for a sequence of basic amino acids (QRKRLKSSCK). Polybasic amino acid stretches and prolines have been reported to cause ribosome stalling (*Ingolia et al., 2011*; *Charneski and Hurst, 2013*; *Brandman et al., 2012*). Since the sequence immediately adjacent to the actual stall site also contained several lysine, arginine, and proline codons, we cloned the stalling sequence with 100 nt upstream and downstream flanking regions to include all potential contributing regulatory elements in the reporter constructs. We then measured the half-lives of the reporter constructs when transfected into HEK293T cells. For this, reporter mRNA was quantified by northern blot at different time points after transcription was blocked using actinomycin D. As expected, the presence of the *AR* stalling sequence before the stop-codon strongly reduced the mRNA half-life compared to the control mRNA where the stop-codon precedes the stalling sequence (*Figure 4B and C*). The same was true for the *BMP2* stalling sequence: the presence of the stalling sequence was associated with a strong translation-dependent reduction in mRNA half-life (4.0±1.9 hr compared to >10 hr in the control vector) (*Figure 4D and E*). To test whether the observed reduction in reporter mRNA stability containing endogenous ribosomal stalling sites was dependent on DDX6, we also performed the half-life measurements in HEK293T DDX6 KO cells, and observed that the R-LUC-stop and R-LUC constructs for both *AR* and *BMP2* stalling sequences had similar half-lives (*Figure 4B–E*). These results, therefore, confirm that DDX6 is required to trigger the co-translational mRNA decay induced by these sequences.

To further confirm that the identified sequences of the *AR* and *BMP2* mRNAs lead to ribosome stalling, we performed RNA pulldown assays of the reporter mRNA in DDX6 KO cells and quantified associated HA-RPL22 levels. In keeping with the mRNA half-life results, the reporters with the *AR* or *BMP2* stalling sequences pulled down increased amounts of HA-RPL22 indicative of increased ribosome occupancy on these reporters (*Figure 4F*).

In summary, using ribosome profiling and follow-up experiments, we were able to identify endogenous mRNA sequences with reduced elongation rates that trigger co-translational mRNA decay in a DDX6-dependent manner in human cells.

## Discussion

mRNA decay factors regulate gene expression by processes other than just general mRNA degradation. Recent studies have shown that the translational elongation rate influences mRNA stability (*Forrest et al., 2020*; *Wu et al., 2019*; *Hia et al., 2019*; *Narula et al., 2019*). One factor implicated in coupling mRNA decay with translation rates is the yeast decay factor, Dhh1p (*Radhakrishnan et al., 2016*). However, this has not been confirmed in human cells yet. In this study, we demonstrated that the inefficiency of translation modulates mRNA stability in human cells, aligning with findings in yeast. Moreover, DDX6 (the human ortholog of Dhh1p) plays an important role in triggering the decay of slowly translated mRNAs. Using a wide range of approaches, we systematically characterized the contribution of DDX6 to ribosome stalling-mediated mRNA decay.

To investigate whether DDX6 is necessary to trigger the decay of sub-optimally translated mRNAs, we created a reporter construct containing a *Renilla* luciferase open reading frame where the last 30 codons were replaced by synonymous but rarely used codons, chosen to reduce the translation rate without altering the amino acid sequence. The reporter with the rare codons was much more stable in the absence of DDX6, with a half-life comparable to that of the corresponding control reporter.

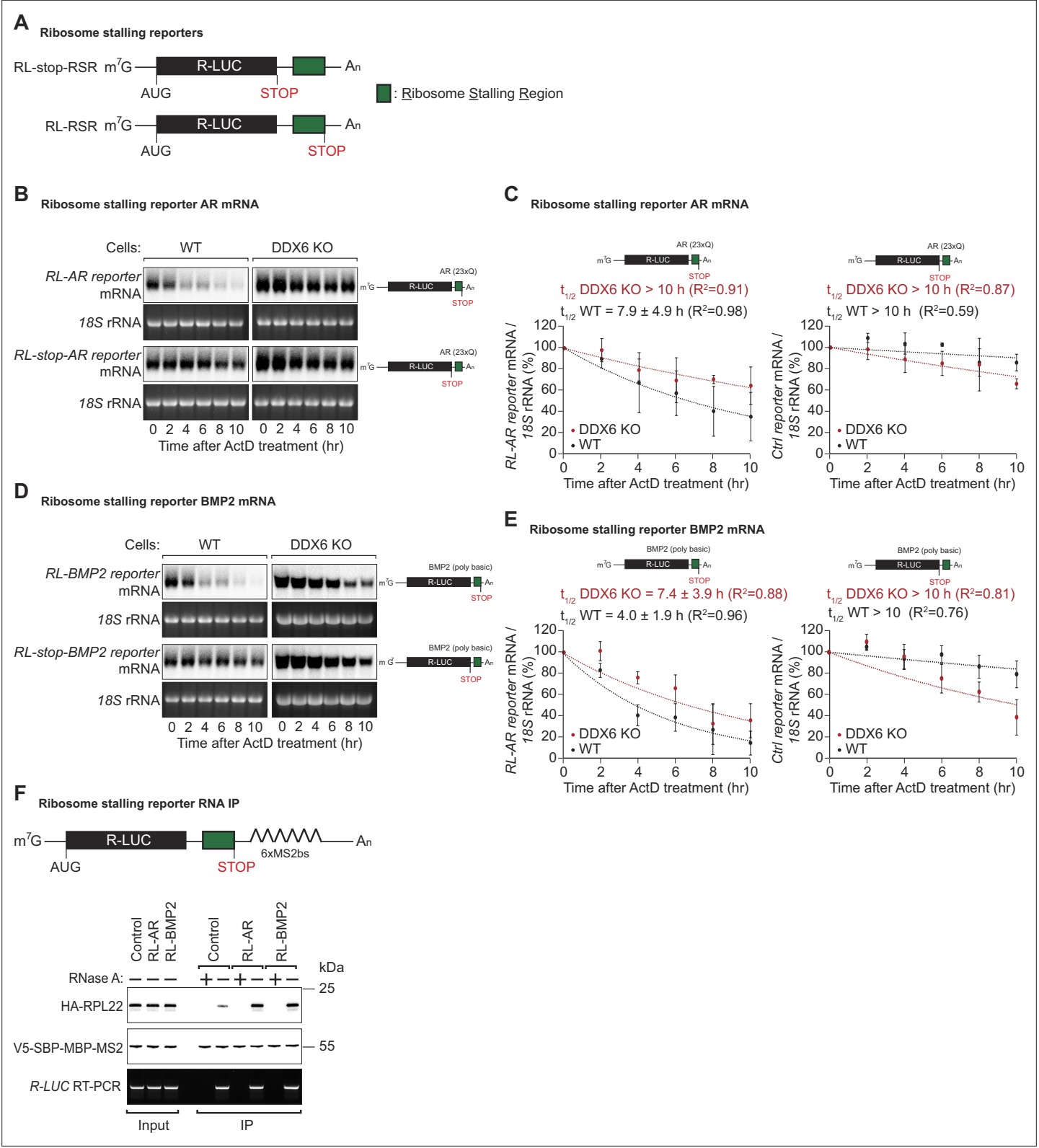

**Figure 4.** DDX6 is required for ribosome-stalling mRNA degradation. (**A**) Schematic representation of the reporters used in panels (**B, C**). (**B**) Representative northern blots showing the decay of androgen receptor (*AR*) reporter mRNAs in HEK293T wild-type (WT) or DDX6 KO cells. Cells were transfected with indicated reporter plasmids and monitored after the inhibition of transcription using actinomycin D (ActD) for the indicated time. *18* S rRNA ethidium bromide staining shows equal loading. (**C**) Relative reporter mRNA levels from panel B at time point zero (before ActD addition)

*Figure 4 continued on next page*

*Figure 4 continued*

were defined as 100%. Relative reporter mRNA levels were plotted as a function of time. Circles represent the mean value and error bars the standard deviation (SD) (n=3). The decay curves were fitted to an exponential decay with a single component (dotted lines). $R^2$ values are indicated for each curve. The half-life of each mRNA in WT and DDX6 KO cells is represented as the mean ± SD. (**D**) Representative northern blots showing the decay of *BMP2* reporter mRNAs in HEK293T WT or DDX6 KO cells. Cells were transfected with indicated reporter plasmids and monitored after the inhibition of transcription using ActD for the indicated time. 18 S rRNA ethidium bromide staining shows equal loading. (**E**) Relative reporter mRNA levels from panel D at time point zero (before ActD addition) were defined as 100%. Relative reporter mRNA levels were plotted as a function of time. Circles represent the mean value and error bars the standard deviation (SD) (n=3). The decay curves were fitted to an exponential decay with a single component (dotted lines). $R^2$ values are indicated for each curve. The half-life of each mRNA in WT and DDX6 KO cells is represented as the mean ± SD. (**F**) HEK293T cells were transfected with indicated R-LUC reporters containing 6xMS2 binding sites, HA-tagged RPL22, and SBP-tagged MBP-MS2 plasmids. RNA bound to V5-SBP-MBP-MS2 was immunoprecipitated with Streptavidin beads. The presence of HA-tagged RPL22 in the immunoprecipitates was determined by western blotting. V5-SBP-MBP-MS2 protein level and RT-PCR of R-LUC reporter RNA levels served as a loading control.

The online version of this article includes the following source data for figure 4:

**Source data 1.** Original file for the northern blot in *Figure 4B*.

**Source data 2.** PDF containing *Figure 4B* and original scans of the relevant northern blot analysis with highlighted bands and sample labels.

**Source data 3.** Original file for the northern blot in *Figure 4D*.

**Source data 4.** PDF containing *Figure 4D* and original scans of the relevant northern blot analysis with highlighted bands and sample labels.

**Source data 5.** Original file for the RNA IP analysis in *Figure 4D*.

**Source data 6.** PDF containing *Figure 4F* and original scans of the relevant RNA IP analysis with highlighted bands and sample labels.

This result illustrated that DDX6 serves as a mediator of ribosome stalling and mRNA decay, and this role is evolutionarily conserved from yeast to mammals. We also demonstrated that DDX6 interacts with the ribosome through the RecA2 domain, which in turn binds to the CCR4-NOT complex. By overexpressing DDX6 mutants in DDX6 KO cells, we showed that DDX6-CCR4-NOT interactions are essential for ribosome stalling-dependent mRNA decay.

To identify a possible link between increased ribosomal occupancy and DDX6-dependent mRNA decay, we performed RNA-seq and ribosome profiling experiments in HEK293T WT and DDX6 KO cells. Comparison of the two RNA-seq libraries allowed us to identify mRNAs that are upregulated in the absence of DDX6. We further examined DDX6-regulated mRNA targets using ribosome profiling and found increased ribosome occupancy in specific regions, indicative of slow ribosome movement, which may, therefore, destabilize the corresponding mRNAs. Importantly, we also found that reporter mRNAs containing the ribosomal stalling region were only destabilized in the presence of both translation and DDX6. These findings have important implications for understanding the regulation of gene expression and the underlying mechanisms of mRNA decay.

While our findings demonstrate a requirement for DDX6 in mRNA decay induced by ribosomal stalling and its interaction with the ribosome, the exact mechanism by which DDX6 may sense ribosome speed remains unclear. Without structural or biochemical data demonstrating recognition of the slowed ribosome by DDX6, the role of DDX6 as a sensor remains one of the possible models. It is conceivable that DDX6 might act as a bridge between the sensor and decay machinery on the ribosome, or it might play a more direct role in sensing ribosome speed. These possibilities present intriguing avenues for future research. By elucidating the endogenous targets, as well as the role of DDX6 and the CCR4-NOT complex in ribosomal stalling-coupled mRNA decay, we may be able to develop more effective therapeutic strategies for diseases caused by dysregulated gene expression. Furthermore, the approach used here of combining RNA-seq and ribosome profiling technologies could be applied to other biological systems to gain a deeper understanding of the molecular mechanisms underlying translation-coupled mRNA decay. Overall, our work opens up exciting avenues for future research and could potentially contribute to the development of novel treatments for a variety of diseases.

## Materials and methods
### DNA constructs

The plasmid for expression of the rare codon reporter pCIneo-RLuc (30xRC) was generated by site-directed mutagenesis using the QuikChange Site-Directed Mutagenesis kit (Stratagene). Protein

mutant plasmids used in this study are listed in *Supplementary file 1*. All constructs were fully sequenced to confirm the presence of the mutations and the absence of additional mutations.

## Cell culture

Human HEK293T cells were purchased from ATCC (CRL-3216) and were cultured in DMEM (Gibco 11995) supplemented with 10% (v/v) fetal bovine serum (Gibco) and grown at 37°C with 5% $CO_2$. The identity of these cells was authenticated through SRT profiling, and they were confirmed to be negative for mycoplasma.

## Generation of a HEK293T DDX6 KO cell line

The knockout of DDX6 in HEK293T cells was generated by CRISPR-Cas9. The guide RNA targeting Exon 3 of DDX6 (5′- GTCTTTTTCCAGTCATCACC –3′) was designed using DNA2.0 (ATUM) online tool to minimize off-target effects. Targeting resulted in a 1 nucleotide insertion in one allele and a 10-nucleotide deletion in the other allele, both causing a frameshift in the open reading frame and severe reduction of *DDX6* mRNA expression (*Figure 1—figure supplement 1B*).

## Northern blotting and quantification

For mRNA half-life measurements, HEK293T wild-type, or DDX6 KO cells were seeded in six-well plates and transfected with 0.5 μg of the indicated *Renilla* luciferase (R-LUC) reporter plasmid. 48 hr after transfection, the R-LUC reporter mRNA level was monitored following inhibition of transcription by actinomycin D. For tethering assays, HEK293T wild-type, or DDX6 KO cells were seeded in six-well plates and transfected with 0.25 μg of the indicated R-LUC reporter plasmid containing 5xBoxB (R-LUC-5BoxB) and 0.25 μg *Firefly* luciferase (F-LUC) plasmid as transfection control, and 1 μg plasmids expressing the $\lambda$ N-HA peptide or $\lambda$ N-HA-CNOT3. For total RNA isolation, cells were resuspended in TRIzol (Thermo Fisher), and RNA was isolated using Phenol:Chloroform: Isoamyl alcohol (PanReac). For northern blotting, 10 μg RNA samples were separated on 2% glyoxal agarose gels and transferred onto nylon membranes (GeneScreen Plus, Perkin Elmer). Complementary [$^{32}$P]-labeled probes were synthesized by linear PCR and hybridized at 65 °C overnight in hybridization solution (0.5 M NaP pH = 7.0, 7% SDS, 1 mM EDTA pH = 8.0). After repeated washing in washing solution (0.5 M NaP pH = 7.0, 1% SDS, 1 mM EDTA pH = 8.0), membranes were scanned and quantified using the Typhoon phosphorimager (GE Healthcare). R-LUC reporter mRNA levels were then normalized to 18 S rRNA and plotted against time. mRNA half-lives ($t_{1/2}$) ± standard deviations were calculated from the decay curves.

## Ribosomal protein purification from HEK293T cells

The method to purify ribosomal proteins from HEK293T cells was adapted from a previously published protocol (*Belin et al., 2010*). In brief, HEK293T cells were submitted to an osmotic shock by incubation in a hypotonic buffer containing NP-40. After removing the nuclear fraction by centrifugation at 750 g, the cytoplasmic fraction was spun down at 12,500 g to remove the mitochondrial fraction. The post-mitochondrial fraction was then adjusted to 0.5 M KCl to disrupt most interactions between ribosomes and other proteins of the other cell compartments. Finally, ribosomes were purified through a sucrose cushion by ultracentrifugation.

## Protein pulldown assay

NusA-Strep-tagged DDX6 was produced in *E. coli* BL21(DE3) Star cells (Thermo Fisher) in LB medium at 20 °C. For the purification, cells were lysed by sonication in PBST buffer. NusA-Strep-tagged DDX6 was isolated from the crude lysate on Strep-Tactin sepharose (IBA) resin for 1 hr at 4 °C. The resin was then washed twice with PBST buffer and once with binding buffer [50 mM Tris/HCl (pH 7.5), 150 mM NaCl]. Purified human ribosome proteins were then added to the resin and incubated for 1 hr at 4 °C. After three washes with the binding buffer, bound proteins were eluted with 2 x protein sample buffer (100 mM Tris-HCl pH = 6.8, 4% SDS, 20% glycerol, 0.2 M DTT) and analyzed by SDS-PAGE followed by Coomassie staining.

## Immunoprecipitation assays and western blotting

HEK293T cells were grown in 10 cm dishes and transfected using Lipofectamine 2000 (Invitrogen). The transfection mixtures contained 10 μg of GFP- and HA-tagged constructs. HEK293T cells were

harvested 48 hr after transfection in NET buffer (50 mM Tris, pH = 7.4, 1 mM EDTA, 150 mM NaCl, 0.1% Triton X-100) supplemented with 10% glycerol and protease inhibitors (Complete Protease Inhibitor Mix, Roche). Cell lysates were cleared by centrifugation at 18,000 g for 15 min and input samples were collected for western blotting. Co-immunoprecipitation experiments were performed in the presence of 1.5 U/µl RNase I (Ambion). Lysates were first incubated with 5 µl anti-GFP antibody (in-house) for 1 hr rotating at 4 °C and subsequently mixed with 50 µl protein G-agarose (50% slurry, Roche) and incubated for 1 hr rotating at 4 °C. After repeated washing in NET buffer, proteins were eluted with 2 x protein sample buffer and analyzed by SDS-PAGE. For western blotting, proteins were transferred onto nitrocellulose membranes (Santa Cruz) via tank transfer and developed with freshly mixed solutionA:solutionB (10:1) and 0.01% $H_2O_2$ [SolutionA: 0.025% Luminol (Roth) in 0.1 M Tris-HCl pH = 8.6; SolutionB: 0.11% P-Coumaric acid (Sigma Aldrich) in DMSO]. HA- and GFP-tagged proteins were detected using horseradish peroxidase-conjugated monoclonal anti-HA (Roche, 3F10, 1:5,000) and anti-GFP (Roche, 11814460001, 1:2,000), respectively. Endogenous human RPS3A was detected using a polyclonal anti-RPS3A antibody (abcam, ab264368, 1:1,000), CNOT1 was detected using a rabbit anti-CNOT1 antibody (in-house, 1:1,000), DDX6 was detected using a rabbit polyclonal anti-DDX6 antibody (Bethyl, A300-461Z, 1:1000), Tubulin was detected using a mouse monoclonal anti-Tubulin antibody (Sigma Aldrich, T6199, 1:3000), and V5-SBP-MBP-MS2 was detected using a mouse monoclonal anti-V5 antibody BioRad, MCA1360GA, 1:5000.

## Reporter-based RNA pulldown assay

HEK293T cells were plated in a 10 cm dish at 80% confluence. Indicated R-LUC reporter, HA-tagged RPL22, and SBP-MBP-MS2 plasmids were transfected into HEK293T cells. After 48 hr, the cells were collected and washed once with pre-cooled PBS. Cells were then lysed in NET buffer (50 mM Tris pH = 7.4, 1 mM EDTA, 150 mM NaCl, 0.1% Triton X-100). The lysate was centrifuged for 5 min at 13,000 g at 4 °C. The supernatant was incubated with Strep-Tactin sepharose (IBA) resin with/without RNase A for 1 hr at 4 °C. After three washes with NET buffer, bound proteins were eluted with 2 x protein sample buffer and analyzed by western blotting.

## RNA-seq and ribosome profiling

For RNA-seq and ribosome profiling experiments HEK293T wild-type or DDX6 KO cells were plated on 15 cm dishes 24 hr before harvesting. Cells were harvested as described in *Calviello et al., 2016*. For total RNA libraries, RNA was extracted using the RNeasy Mini Kit (Qiagen) and processed according to the Illumina TruSeq RNA Sample Prep Kit. For ribosome profiling the original protocol (*Ingolia et al., 2012*) was used in a modified version described in *Calviello et al., 2016*.

RNA-seq and ribosome profiling libraries were sequenced on an Illumina Hiseq3000 instrument. Reads originating from ribosomal RNA were removed using Bowtie2 (*Langmead and Salzberg, 2012*). Remaining reads were mapped onto the human genome using Tophat2 (*Kim et al., 2013*) which resulted in 21–27 million mapped reads with an overall read mapping rate >91% for RNA-seq and 3–4 million mapped reads with an overall read mapping rate >87% for ribosome profiling datasets. Read count analysis was performed using QuasR (*Gaidatzis et al., 2015*), and differential expression analysis was performed using edgeR (*Robinson et al., 2010*; *McCarthy et al., 2012*). Translational efficiencies (TE) were calculated using RiboDiff (*Zhong et al., 2017*).

DDX6 stalling-induced decay targets were selected I. based on the logFC of total RNA of DDX6 KO versus WT HEK293T cells. Targets with a logFC >1 and statistically significant change FDR <0.005 (n=298) were manually screened for potential stalling sites using the Integrative Genomics Viewer tool from Broad Institute (*Robinson et al., 2011*; *Thorvaldsdóttir et al., 2013*).

II. Stalling targets were selected based on a local accumulation of footprints determined by ribosome profiling. We found 35 genes that matched these criteria (AMDHD2, AR, ART5, BMP2, C1QTNF4, CA11, CALCB, CD19, CILP2, COL2A1, CTSA, CXCL12, DLX5, DUSP1, ENO2, HOXA13, IFI35, IFITM2, IGFBP5, LGALS1, MLC1, PBXIP1, PCK1, PSMB9, PTRH1, QPCT, SAT1, SEMA6C, SFXN3, SLC13A4, SLIT2, SMPDL3B, TMSB15B, TNC, ZNF91). A representation of the most convincing targets (AR, BMP2, LGALS1, DLX5, ENO2, PSMB9) can be found in *Figure 3—figure supplement 2B–G*.

GO analysis was performed using the R-based package goseq (*Young et al., 2010*). Analyses of GC content, CDS length, and transcript TE were performed with R-based scripts.

## Reverse transcription (RT) and quantitative PCR (qPCR)

For quantitative PCR (qPCR) of selected targets we designed 22 nt primers using Primer-BLAST from NCBI to amplify a 250–350 bp region from cDNA. To use the Livak method (ΔΔCT method) for relative quantification each primer pair was tested if the amplification rate of the specific PCR product is 2+/-5%.

For validation and rescue experiments HEK293T wild-type or DDX6 KO cells were either transfected with a GFP-MBP control or GFP-DDX6 (either WT, Mut1, Mut2, DEAD*). 48 hr after transfection total RNA was extracted and reverse transcribed using random hexamer primers. Target mRNA amount was subsequently determined by qPCR using the respective primer pair and normalized to beta-Actin abundance in the same sample. A sample treated with water instead of reverse transcriptase (-RT) was used as a negative control. Relative abundances of target mRNAs in three independent experiments were determined using the Livak method (the wild-type sample was arbitrarily set to 1) (*Livak and Schmittgen, 2001*).

## Acknowledgements

We dedicate this work to the memory of Elisa Izaurralde who passed away while this work was still ongoing. We are grateful to Lara Wohlbold for her contributions to project administration, investigation, and assistance in drafting the initial manuscript during the early stages of the project. We thank Catrin Weiler, Sigrun Helms, and Heike Budde for their excellent technical assistance. We are very grateful to Cátia Igreja and Eugene Valkov for their insightful comments on the project.

## Additional information

### Funding

| Funder | Grant reference number | Author |
| --- | --- | --- |
| National Science and Technology Council | NSTC111-2311-B-A49-005- | Chung-Te Chang |
| Yen Tjing Ling Medical Foundation | CI-112-13 | Chung-Te Chang |

The funders had no role in study design, data collection and interpretation, or the decision to submit the work for publication. Open access funding provided by Max Planck Society.

### Author contributions

Ramona Weber, Conceptualization, Resources, Data curation, Software, Formal analysis, Validation, Investigation, Visualization, Methodology, Writing - original draft, Project administration, Writing - review and editing; Chung-Te Chang, Conceptualization, Resources, Data curation, Supervision, Funding acquisition, Validation, Investigation, Methodology, Writing - original draft, Project administration, Writing - review and editing

### Author ORCIDs

Ramona Weber http://orcid.org/0000-0003-4176-1685
Chung-Te Chang http://orcid.org/0000-0002-4792-1646

Reviewer #1 (Public Review): https://doi.org/10.7554/eLife.92426.3.sa1
Reviewer #2 (Public Review): https://doi.org/10.7554/eLife.92426.3.sa2
Author response https://doi.org/10.7554/eLife.92426.3.sa3

## Additional files

### Supplementary files

- MDAR checklist

• Supplementary file 1. Constructs and mutants used in this study.

## Data availability

The datasets generated in this study have been deposited in the NCBI Gene Expression Omnibus (GEO) with the accession number GSE231964.

The following dataset was generated:

| Author(s) | Year | Dataset title | Dataset URL | Database and Identifier |
|---|---|---|---|---|
| Weber R, Chang C | 2024 | Human DDX6 regulates translation and decay of inefficiently translated mRNAs | https://www.ncbi.nlm.nih.gov/geo/query/acc.cgi?acc=GSE231964 | NCBI Gene Expression Omnibus, GSE231964 |

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

# Appendix 1

**Appendix 1—key resources table**

| Reagent type (species) or resource | Designation | Source or reference | Identifiers | Additional information |
|---|---|---|---|---|
| Gene (*Homo sapiens*) | DDX6 | GenBank | HGNC:2747 | |
| Strain, strain background (*Escherichia coli*) | BL21 Star (DE3) | Thermo Fisher | Invitrogen: C601003 | |
| Cell line (*H. sapiens*) | HEK293T | ATCC | CRL-3216 | Identity authenticated by SRT profiling, negative for mycoplasma. |
| Cell line (*H. sapiens*) | HEK293T DDX6 KO | Elisa Izaurralde Lab | *Hanet et al., 2019* | Developed and maintained by Elisa Izaurralde lab, identity authenticated by SRT profiling, negative for mycoplasma. This material can be obtained from the Elisa Izaurralde Lab. |
| Transfected construct (*E. coli*) | pnEK-NvHM-Strep-MBP | Elisa Izaurralde Lab | *Chang et al., 2019* | This material can be obtained from the Elisa Izaurralde Lab. |
| Transfected construct (*E. coli*) | pETM-60-NusA-3C-HsRCK_296–472-Strep | Elisa Izaurralde Lab | Addgene #146209 | Addgene #146209 |
| Transfected construct (*H. sapiens*) | pT7-EGFP-C1-MBP | Elisa Izaurralde Lab | Addgene #146318 | Addgene #146318 |
| Transfected construct (*H. sapiens*) | pT7-EGFP-C1-HsDDX6 | Elisa Izaurralde Lab | Addgene #25033 | Addgene #25033 |
| Transfected construct (*H. sapiens*) | pT7-EGFP-C1-HsDDX6_1–295 | Elisa Izaurralde Lab | This paper | This material can be obtained from the Elisa Izaurralde Lab. |
| Transfected construct (*H. sapiens*) | pT7-EGFP-C1-HsDDX6_296–463 | Elisa Izaurralde Lab | Addgene #145971 | Addgene #145971 |
| Transfected construct (*H. sapiens*) | pT7-EGFP-C1-HsDDX6_E236Q | Elisa Izaurralde Lab | Addgene #146456 | Addgene #146456 |
| Transfected construct (*H. sapiens*) | pT7-EGFP-C1-HsDDX6_Mut1 | Elisa Izaurralde Lab | Addgene #147023 | Addgene #147023 |
| Transfected construct (*H. sapiens*) | pT7-EGFP-C1-HsDDX6_Mut2 | Elisa Izaurralde Lab | Addgene #148452 | Addgene #148452 |
| Transfected construct (*H. sapiens*) | pCIneo-HA-RPL22 | Elisa Izaurralde Lab | This paper | This material can be obtained from the Elisa Izaurralde Lab. |
| Transfected construct (*H. sapiens*) | pCIneo-RLuc | Elisa Izaurralde Lab | Addgene #146090 | Addgene #146090 |
| Transfected construct (*H. sapiens*) | pCIneo-RLuc_ 30xRC | Elisa Izaurralde Lab | This paper | This material can be obtained from the Elisa Izaurralde Lab. |
| Transfected construct (*H. sapiens*) | pCIneo-RL-AR | Elisa Izaurralde Lab | This paper | This material can be obtained from the Elisa Izaurralde Lab. |
| Transfected construct (*H. sapiens*) | pCIneo-RL-Stop-AR | Elisa Izaurralde Lab | This paper | This material can be obtained from the Elisa Izaurralde Lab. |
| Transfected construct (*H. sapiens*) | pCIneo-RL-BMP2 | Elisa Izaurralde Lab | This paper | This material can be obtained from the Elisa Izaurralde Lab. |
| Transfected construct (*H. sapiens*) | pCIneo-RL-Stop-BMP2 | Elisa Izaurralde Lab | This paper | This material can be obtained from the Elisa Izaurralde Lab. |
| Transfected construct (*H. sapiens*) | pCIneo-v5-SBP-MBP-MS2 | Elisa Izaurralde Lab | This paper | This material can be obtained from the Elisa Izaurralde Lab. |
| Transfected construct (*H. sapiens*) | pCIneo-RL-6xMS2bs | Elisa Izaurralde Lab | Addgene #148306 | Addgene #148306 |
| Transfected construct (*H. sapiens*) | pCIneo-RL-AR-6xMS2bs | Elisa Izaurralde Lab | This paper | This material can be obtained from the Elisa Izaurralde Lab. |

*Appendix 1 Continued on next page*

*Appendix 1 Continued*

| Reagent type (species) or resource | Designation | Source or reference | Identifiers | Additional information |
|---|---|---|---|---|
| Transfected construct (*H. sapiens*) | pCIneo-RL-BMP2-6xMS2bs | Elisa Izaurralde Lab | This paper | This material can be obtained from the Elisa Izaurralde Lab. |
| Antibody | anti-GFP (Rabbit polyclonal) | Elisa Izaurralde Lab | *Chen et al., 2014* | IP (This material can be obtained from the Elisa Izaurralde Lab.) |
| Antibody | anti-GFP (Mouse monoclonal) | Roche | Roche #11814460001 | WB(1:2000) |
| Antibody | anti-HA-HRP (Mouse monoclonal) | Roche | Roche #12013819001 | WB(1:5000) |
| Antibody | anti-CNOT1 (Rabbit polyclonal) | Elisa Izaurralde Lab | *Chen et al., 2014* | WB(1:1000) |
| Antibody | anti-DDX6 (Rabbit polyclonal) | Bethyl, A300-461Z | Bethyl #A300-461Z | WB(1:1000) |
| Antibody | anti-RPS3A (Rabbit polyclonal) | Abcam | Abcam #ab264368 | WB(1:1000) |
| Antibody | anti-V5 (Mouse monoclonal) | BioRad | BioRad #MCA1360GA | WB(1:5000) |
| Antibody | anti-Tubulin (Mouse monoclonal) | Sigma Aldrich | Sigma Aldrich #T6199 | WB(1:3000) |
| Sequence-based reagent | AR_F | This paper | qPCR primers | gacatgcgtttggagactgcca |
| Sequence-based reagent | AR_R | This paper | qPCR primers | cccagagtcatccctgcttcat |
| Sequence-based reagent | BMP2_F | This paper | qPCR primers | cccagagtcatccctgcttcat |
| Sequence-based reagent | BMP2_R | This paper | qPCR primers | cagcaacgctagaagacagcgg |
| Sequence-based reagent | LGALS1_F | This paper | qPCR primers | ctcaaacctggagagtgccttc |
| Sequence-based reagent | LGALS1_R | This paper | qPCR primers | tcgtatccatctggcagcttga |
| Sequence-based reagent | PSMB9_F | This paper | qPCR primers | cttttgccattggtggctccgg |
| Sequence-based reagent | PSMB9_R | This paper | qPCR primers | ccataccaggttttggccctag |
| Sequence-based reagent | GAPDH_F | This paper | qPCR primers | ctctgctcctcctgttcgacag |
| Sequence-based reagent | GAPDH_R | This paper | qPCR primers | ttcccgttctcagccttgacgg |
| Sequence-based reagent | Beta-Actin_F | This paper | qPCR primers | ccaaaagcatgacaggcagaaa |
| Sequence-based reagent | Beta-Actin_R | This paper | qPCR primers | tcccgtgttcctcaccaatcat |
| Sequence-based reagent | DLX5_F | This paper | qPCR primers | CAGCCATGTCTGCTTAGACCAG |
| Sequence-based reagent | DLX5_R | This paper | qPCR primers | TACTGGTAGGGGTTGAGAGCTT |
| Sequence-based reagent | ENO2_F | This paper | qPCR primers | ATGTGTCACTTGTGCTTTGCTC |
| Sequence-based reagent | ENO2_R | This paper | qPCR primers | ACCCCAGTCATCTTGGGATCTA |
| Commercial assay or kit | RNeasy Mini Kit | Qiagen | Qiagen 74104 | |
| Commercial assay or kit | TruSeq RNA Library Prep Kit v2 | Illumina | Illumina RS-122–2002 | |
| Commercial assay or kit | Ribo-Zero Gold Kit | Illumina | discontinued | |
| Chemical compound, drug | Actinomycin D | Sigma-Aldrich | Sigma-Aldrich #A1410 | |
| Software, algorithm | Bowtie2 | *Langmead and Salzberg, 2012* | | |
| Software, algorithm | Tophat2 | *Kim et al., 2013* | | |

*Appendix 1 Continued on next page*

*Appendix 1 Continued*

| Reagent type (species) or resource | Designation | Source or reference | Identifiers | Additional information |
|---|---|---|---|---|
| Software, algorithm | QuasR | *Gaidatzis et al., 2015* | | |
| Software, algorithm | edgeR | *Robinson et al., 2010*; *McCarthy et al., 2012* | | |
| Software, algorithm | RiboDiff | *Zhong et al., 2017* | | |
| Software, algorithm | Integrative Genomics Viewer (IGV) | Broad Institute; *Robinson et al., 2011*; *Thorvaldsdóttir et al., 2013* | | |
| Software, algorithm | goseq | *Young et al., 2010* | | |

