## [Editor Report · eLife assessment]

This study provides **valuable** findings that improve our understanding of the evolutionary conservation of the role of DDX6 in mRNA decay. The evidence supporting the authors' conclusions is **convincing**. This work will be of interest to molecular, cell biologists and biochemists, especially those studying RNA.

---

## [Referee Report · Reviewer #1 (Public Review)]

Weber et al. investigated the role of human DDX6 in messenger RNA decay using CRISPR/Cas9 mediated knockout (KO) HEK293T cells. The authors showed that stretches of rare codons or codons known to cause ribosome stalling in reporter mRNAs leads to a DDX6 specific loss of mRNA decay. The authors moved on to show that there is a physical interaction between DDX6 and the ribosome. Using co-immunoprecipitation (co-IP) experiments, the authors determined that the FDF-binding surface of DDX6 is necessary for binding to the ribosome, the same domain which is necessary for binding several decapping factors such as EDC3, LSM14A, and PatL. However, they determine the interaction between DDX6, and the ribosome is independent of the DDX6 interaction with the NOT1 subunit of the CCR4-NOT complex. Interestingly, the authors were able to determine that all known functional domains, including the ATPase activity of DDX6, are required for its effect on mRNA decay. Using ribosome profiling and RNA-sequencing, the authors were able to identify a group of 260 mRNAs that exhibit increased translational efficiency (TE) in DDX6 Knockout cells, suggesting that DDX6 translationally represses certain mRNAs. The authors determined this group of mRNAs has decreased GC content, which has been previously noted to coincide with low codon optimality, the authors thus conclude DDX6 may translationally repress transcripts of low codon optimality. Furthermore, the authors identify 35 transcripts that are both upregulated in DDX6 KO cells and exhibit locally increased ribosome footprints (RBFs), suggestive of a ribosome stalling sequence. Lastly, the authors showed that both endogenous and tethering of DDX6 to reporter mRNAs with and without these translational stalling sequences leads to a relative increase in ribosome association to a transcript. Overall, this work confirms that the role of DDX6 in mRNA decay shares several conserved features with the yeast homolog Dhh1. Dhh1 is known to bind slow-moving ribosomes and lead to the differential decay of non-optimal mRNA transcripts (Radhakrishnan et al. 2016). The novelty of this work lies primarily in the identification of the physical interaction between DDX6 and the ribosome and the breakdown of which domains of DDX6 are necessary for this interaction. This work provides major insight into the role of the human DDX6 in the process of mRNA decay and emphasizes the evolutionary conservation of this process across Eukaryotes.

Overall, the work done by Weber et al. is sound, with the proper controls. The authors expand significantly on the knowledge of what we know about DDX6 in the process of mRNA decay in humans, confirming the evolutionary conservation of the role of this factor across eukaryotes. The analysis of the RNA-seq and Ribo-seq data could be more in-depth, however, the authors were able to show with certainty that some transcripts containing known repetitive sequences or polybasic sequences exhibited a DDX6-mRNA decay effect.

---

## [Referee Report · Reviewer #2 (Public Review)]

In the manuscript by Weber and colleagues, the authors investigated the role of a DEAD-box helicase DDX6 in regulating mRNA stability upon ribosome slowdown in human cells. The authors knocked out DDX6 KO in HEK293T cells and showed that the half-life of a reporter containing a rare codon repeat is elongated in the absence of DDX6. By analogy to the proposed function of fission yeast Dhh1p (DDX6 homolog) as a sensor for slow ribosomes, the authors demonstrated that recombinant DDX6 interacted with human ribosomes. The interaction with the ribosome was mediated by the FDF motif of DDX6 located in its RecA2 domain, and rescue experiments showed that DDX6 requires the FDF motif as well as its interaction with the CCR4-NOT deadenylase complex and ATPase activity for degrading a reporter mRNA with rare codons. To identify endogenous mRNAs regulated by DDX6, they performed RNA-Seq and ribosome footprint profiling. The authors focused on mRNAs whose stability is increased in DDX6 KO cells with high local ribosome density and validated that such mRNA sequences induced mRNA degradation in a DDX6-dependent manner.

The experiments were well-performed, and the results clearly demonstrated the requirement of DDX6 in mRNA degradation induced by slowed ribosomes.

[Editors' note: The authors have addressed the key points from the previous public reviews in their revised manuscript.]

---

## [Author Response]

The following is the authors’ response to the original reviews.

**Public Reviews:**

**Reviewer #1 (Public Review):**
Weaknesses:The authors fail to truly define codon optimality, rare codons, and stalling sequences in their work, all of which are distinct terminologies. They use reporters with rare codon usage but do not mention what metrics they use to determine this, such as cAI, codon usage bias, or tAI. The distinction between the type of codon sequences that DDX6 affects is very important to differentiate and should be done here as certain stretches of codons are known to lead to different quality control RNA decay pathways that are not reliant on canonical mRNA decay factors.

Thank you for the reviewer’s feedback on our work. Clearly defining codon optimality, rare codons, and stalling sequences is indeed crucial. We will emphasize this distinction more in our revisions to help readers better understand our analysis and findings.

Likewise, the authors sort their Ribo-seq data to determine genes that might exhibit a DDX6 specific mRNA decay effect but fail to go into great depth about common features shared among these genes other than GO term analysis, GC content, and coding sequence (CDS) length. The authors then sort out 35 genes that are both upregulated at the mRNA level and have increased local ribosome footprint along the ORF. They are then able to show that 6 out of 9 of those genes had a DDX6-dependent mRNA decay effect. There was no comment or effort as to why 2 out of those 6 genes tested did not show as strong of a DDX6-dependent decay effect relative to the other targets tested. Thus, the efforts to identify mRNA features at a global level that exhibited DDX6-dependent mRNA decay effects are lacking in this analysis.

We appreciate the reviewer's insightful comments regarding the need to further characterize the genes influenced by DDX6-mediated mRNA decay. To address this, we carried out additional analyses to identify potential traits of these genes. Our findings revealed that DDX6-regulated coding sequences tend to be longer and exhibit lower predicted mRNA stability scores compared to the average across the transcriptome. This observation indicates a possible connection to codon optimality. It suggests that DDX6 could play a role in regulating a specific subset of mRNAs with inherently lower stability, potentially shedding light on why some genes may exhibit varied decay patterns when DDX6 is depleted.

Overall, the work done by Weber et al. is sound, with the proper controls. The authors expand significantly on the knowledge of what we know about DDX6 in the process of mRNA decay in humans, confirming the evolutionary conservation of the role of this factor across eukaryotes. The analysis of the RNA-seq and Ribo-seq data could be more in-depth, however, the authors were able to show with certainty that some transcripts containing known repetitive sequences or polybasic sequences exhibited a DDX6-mRNA decay effect.

We appreciate the reviewer’s acknowledgment of the soundness of our work and the inclusion of proper controls. We are committed to refining our manuscript to meet your expectations and ensure the accuracy and depth of our findings.

**Reviewer #2 (Public Review):**
The experiments were well-performed, and the results clearly demonstrated the requirement of DDX6 in mRNA degradation induced by slowed ribosomes. However, in some cases, the authors interpreted their data in a biased way, possibly influenced by the yeast study, and drew too strong conclusions. In addition, the authors should have cited important studies about codon optimality in mammalian cells. This lack of information hinders placing their important discoveries in a correct context.(1) Although the authors concluded that DDX6 acts as a sensor of the slowed ribosome, it is not clear if DDX6 indeed senses the ribosome speed. What the authors showed is a requirement of DDX6 for mRNA decay induced by rare codons, and DDX6 binds to the ribosome to exert this role. For example, DDX6 may bridge the sensor and decay machinery on the ribosome. Without structural or biochemical data on the recognition of the slowed ribosome by DDX6, the role of DDX6 as a sensor remains one of the possible models. It should be described in the discussion section.

We greatly appreciate the reviewer’s comments and suggestions. We agree that our study does not directly establish that DDX6 senses ribosome speed. We also agree that without structural or biochemical data demonstrating recognition of the slowed ribosome by DDX6, the role of DDX6 as a sensor remains one of the possible models. We will incorporate this point into the discussion section and acknowledge it as an important direction for future research.

(2) It is not clear if DDX6 directly binds the ribosome. The authors used ribosomes purified by sucrose cushion, but ribosome-associating and FDF motif-interacting factors might remain on ribosomes, even after RNaseI treatment. Without structural or biochemical data of the direct interaction between the ribosome and DDX6, the authors should avoid description as if DDX6 directly binds to the ribosome.

We agree with the reviewer’s perspective that, even after RNase I treatment, factors associated with the ribosome and interacting with the FDF motif might still remain on the ribosomes that were purified via a sucrose cushion. In the revised manuscript, we will describe the relationship between DDX6 and the ribosome more cautiously, avoiding the depiction of DDX6 directly binding to the ribosome.

(3) Although the authors performed rigorous reporter assays recapitulating the effect of ribosome-retardation sequences on mRNA stability, this is not the first report showing that codon optimality determines mRNA stability in human cells. The authors did not cite important previous studies, such as Wu et al., 2019 (PMID: 31012849), Hia et al., 2019 (PMID: 31482640), Narula et al., 2019 (PMID: 31527111), and Forrest et al., 2020 (PMID: 32053646). These milestone papers should be cited in the Introduction, Results, and Discussion.

Thank you for the reviewer’s correction. We apologize for the oversight in our references. In the revised manuscript, we will ensure these key studies are appropriately cited.

(4) While both DDX6 and deadenylation by the CCR4-NOT were required for mRNA decay by the slowed ribosome, whether DDX6 is required for deadenylation was not investigated. Given that the CCR4-NOT deadenylate complex directly interacts with the empty ribosome E-site in yeast and humans (Buschauer et al., 2020 PMID: 32299921 and Absmeier et al., 2023 PMID: 37653243), whether the loss of DDX6 also affected the action of the CCR4-NOT complex is an important point to investigate, or at least should be discussed in this paper.

We sincerely appreciate the reviewer's valuable suggestions. This point is indeed crucial, and we have addressed it in the revised version of our manuscript. We have included experimental results confirming that the knockout of DDX6 does not impact the CCR4-NOT complex’s deadenylation function. This addition will contribute to a more comprehensive discussion of the relevant issues and refine our manuscript.

**Recommendations for the authors:**

**Reviewer #1 (Recommendations For The Authors):**
The authors should explain what they use to determine rare codons in their system and distinguish this feature from codon optimality. Codon optimality is a distinct feature from rare codon usage, and both should be defined better in the context of the paper. The authors interchange between the use of codon optimality, rare codon usage, and translation stalling sequences frequently and should explain and clarify these terms or consider only referring to translation stalling sequences for their discussion.

We appreciate the reviewer's valuable feedback, we have been able to improve the clarity and rigor of the relevant statements in the manuscript. In the revised manuscript, we have provided more explicit and detailed explanations regarding the definition and use of rare codons, and differentiated this from codon optimality, in order to help readers better understand the basis of our analysis and research findings. Furthermore, in the revised manuscript, we are now referring exclusively to 'translation stalling sequences' in our discussion, in order to provide greater clarity.

**Reviewer #2 (Recommendations For The Authors):**
Interestingly, the translation efficiency of zinc-finger domain mRNAs was increased in DDX6 KO cells. This finding is consistent with the previous study reporting that mRNAs encoding zinc-finger domains are enriched with non-optimal codons and unstable. (Diez et al., 2022 PMID: 35840631). The authors might want to cite this paper and mention the consistency of the two studies.

Thank you for noting the relevance of the increased translation efficiency of zinc-finger domain mRNAs in DDX6 KO cells. We will reference the study by Diez et al. (2022) and emphasize the consistency between their findings and ours, which supports the idea that DDX6 is involved in regulating the translation of mRNAs with these characteristics.

A mutagenesis analysis of the poly-basic residues of BMP2 would further strengthen the authors' claim that this sequence is a primal cause of ribosome slowdown and mRNA decay.

We greatly appreciate the reviewer’s suggestion to conduct a mutagenesis analysis of the poly-basic residues of BMP2. We agree that such an analysis could potentially strengthen our claim. However, considering the constraints we are currently encountering, and our study has already provided substantial evidence to support our findings, we believe that at this stage of our research, conducting this analysis may not be the most immediate priority. We will consider undertaking a mutagenesis analysis in future studies to further validate our conclusions.

In the Introduction, RQC is not commonly referred to as "ribosome-based quality control." Please consider the use of "ribosome-associated quality control."

We appreciate the reviewer providing this suggestion. During the revision process, we corrected the relevant terminology to ensure more precise and appropriate usage.

In the Introduction, the authors should avoid introducing NMD as a part of RQC. NMD was discovered and defined independently of RQC.

Thank you for pointing out this important distinction. We recognize that NMD was discovered and defined independently from RQC, and should not be presented as an integral part of the RQC process. In the revised manuscript, we have made sure to avoid introducing nonsense-mediated decay (NMD) as a component of ribosome-associated quality control (RQC).